# Distribution of Embryonic Stem Cell-Derived Mesenchymal Stem Cells after Intravenous Infusion in Hypoxic–Ischemic Encephalopathy

**DOI:** 10.3390/life13010227

**Published:** 2023-01-13

**Authors:** Su Hyun Lee, Jin Seung Choung, Jong Moon Kim, Hyunjin Kim, MinYoung Kim

**Affiliations:** 1School of Medicine, CHA University, Pocheon 13496, Republic of Korea; 2Rehabilitation and Regeneration Research Center, CHA University, Seongnam 13488, Republic of Korea; 3Department of Rehabilitation Medicine, CHA Bundang Medical Center, CHA University, Seongnam 13488, Republic of Korea; 4Department of Biomedical Science, CHA University, Seongnam 13488, Republic of Korea

**Keywords:** cell tracking, in vivo imaging, hypoxic–ischemic encephalopathy, human embryonic stem Cell-Derived

## Abstract

Systemic administration of mesenchymal stem cells (MSCs) has been reported to improve neurological function in brain damage, including hypoxic–ischemic encephalopathy (HIE), though the action mechanisms have not been fully elucidated. In this study, the cells were tracked live using a Pearl Trilogy Small Animal fluorescence imaging system after human embryonic stem Cell-Derived MSCs (ES-MSCs) infusion for an HIE mouse model. ES-MSC–treated HIE mice showed neurobehavioral improvement. In vivo imaging showed similar sequential migration of ES-MSCs from lungs, liver, and spleen within 7 days in both HIE and normal mice with the exception of lungs, where there was higher entrapment in the HIE 1 h after infusion. In addition, ex vivo experiments confirmed time-dependent infiltration of ES-MSCs into the organs, with similar findings in vivo, although lungs and brain revealed small differences. ES-MSCs seemed to remain in the brain only in the case of HIE on day 14 after the cell infusion. The homing effect in the host brain was confirmed with immunofluorescence staining, which showed that grafted cells remained in the brain tissue at the lesion area with neurorestorative findings. Further research should be carried out to elucidate the role of each host organ’s therapeutic effects when stem cells are systemically introduced.

## 1. Introduction

Cerebral ischemia is a disease in which oxygen is not sufficiently supplied to the brain tissue, resulting in damage to the brain neurons and causing serious disability [1,2]. To investigate the pathological mechanism and therapeutic approaches, hypoxic–ischemic encephalopathy (HIE) models have been used [3,4]. Due to limitations of neuronal recovery in HIE, stem cell–based therapy has emerged for regenerative purposes. Stem cells have the potential of not only self-renewal but also multi-lineage differentiation, which is expected to generate various functional cells and replace the lost tissues [5,6]. In some studies, transplanted cells reportedly reconstituted neural circuits with host cells [7,8]. Moreover, introduced stem cells promote endogenous neurogenesis and survival of viable cells by secreting neurotrophic factors [9]. Accordingly, stem cell therapy can be suggested as a robust method for addressing neuronal damage (e.g., in the case of cerebral ischemia) and may be tested using an HIE model.

Among the stem cell sources, mesenchymal stem cells (MSCs) have been studied for many years, and their immunomodulatory and neuroregenerative characteristics have a demonstrated effect on neurogenesis and neurobehavioral recovery in ischemic brain injury [10,11,12]. In particular, embryonic stem Cell-Derived MSCs (ES-MSCs) have shown a higher potential for neuroprotection, proliferation, and differentiation than other MSCs. Therefore, the cells were investigated for the treatment of brain disease models [11,13]. While some studies have confirmed the efficacy of MSCs, the exact treatment mechanism is still unknown [14,15]. Although the results of early studies on therapeutic cells have suggested that the mechanism involves regeneration through direct replacement in the injured lesions [16], recent research suggests other mechanisms that may exert an effect on the host brain, such as neurotrophic factor secretions, neurogenesis, angiogenesis, and antiapoptosis [17,18]. Among these, systemic reactions, including immune response, have been suggested to be involved in the mechanism [13,14,15], though the mechanism is not yet fully disclosed. Since many reports on stem cell therapy have shown the efficacy of systemically administered MSCs in the case of brain damage, identifying the distribution of the introduced cells would help us understand the therapeutic mechanism

Increasing resolution densities of imaging technologies coupled with advances in cell therapy induced a revolution in cell tracking in vivo [19]. Tracking administered cells using imaging tools in animal models to visualize cell fate, migration among the organs inside the host body, and homing will be useful for cell-based therapy research [20,21]. As a technical source of biomedical imaging, near-infrared (NIR) fluorescence has various advantages, including relatively low tissue absorption, low scattering, and minimal autofluorescence. Unlike visible light, which penetrates biological tissue to a depth of less than 1 mm, NIR light can penetrate biological tissue to depths of millimeters to centimeters [22]. In a previous NIR study, the migration of cells systemically through the vascular pathway was reported. The cells moved from the initial injection site to the target organs, such as gastrointestinal tissue, kidneys, lungs, liver, thymus, and skin [23]. When investigating the biodistribution of stem cells by using NIR, a comparison with the results of ex vivo imaging would be essential for confirming organ specificity, since there may exist influencing factors, such as vasculature and the blood–brain barrier (BBB) [24,25,26]. In this study, the authors purposed to trace the biodistribution of ES-MSCs following the administration of ES-MSCs via the intravenous (IV) route in an HIE model as a basic step to understanding its therapeutic mechanism. Additionally, in vivo NIR activity visualizing the infiltration of the stem cells was compared with the excised organ, both of which showed migration of the introduced ES-MSCs.

## 2. Materials and Methods

### 2.1. Animals

All methods complied with the Animal Research: Reporting of In Vivo Experiments (ARRIVE) guidelines (PLoS Bio 8(6), e1000412, 2010) and other relevant guidelines. All interventions and animal care procedures were performed in accordance with the Laboratory Animal Welfare Act, the Guide for the Care and Use of Laboratory Animals, and the Guidelines and Policies for Animal Surgery provided by the study institute, CHA University, and were approved by the Institutional Animal Care and Use Committee (IACUC 200044). C57BL/6 male mice (6 weeks old, 20–25 g) were purchased from Orient Bio Inc. (Seoul, Republic of Korea). All mice were individually caged and had free access to food and water. The animals were maintained in a temperature-controlled room (23 °C ± 2 °C) under a 12/12-h light/dark cycle (lights on at 08:00; lights off at 20:00).

### 2.2. ES-MSC Labeling with CellVue^®^ NIR815 Fluorophore

Embryonic stem cell (CHA-15)–derived mesenchymal stem cells (ES-MSCs) [27] were labeled with CellVue^®^ NIR815 fluorophore using CellVue^®^ NIR815 Fluorescent Cell Labeling Kit as per the manufacturer’s instructions (Thermofisher Scientific, Waltham, MA, USA). CellVue^®^ NIR815 fluorophore enables detection of the cells on the 800 nm channel by labeling the lipid regions in the cell membrane [28]. The final concentration of the label was 2 μM for 1 × 10^6^ cell/mL density of ES-MSCs, which were placed for 3 min in a serum-free medium. After confirming a viability of >95%, they were washed with phosphate-buffered saline (PBS) and used for in vivo injection.

### 2.3. Hypoxic–Ischemic Brain Injury and the Administration of ES-MSCs

After 5 days of acclimation, the C57Bl/6 mice were anesthetized with isoflurane and the right common carotid artery was ligated with a 5–0 nylon suture. Isoflurane induction, using the Vevo Compact Anesthesia System, was performed by the same individual every time, with the nose of the mouse placed into a small nose cone using 3% isoflurane in pure medical oxygen. Then, the mice were moved to a warmed chamber (37 °C) and exposed to a hypoxic environment (8% O2, 92% N2, and flow rate 1 L/min) for 90 min. Twenty-four hours after this hypoxic–ischemic induction, ES-MSCs labeled with CellVue^®^ (3 × 10^5^ cells/300 μL in PBS) were injected through the tail vein.

### 2.4. Experimental Design

In this study, to investigate the effect of stem cell injection on behavioral improvement in the HIE model, we established an optimal experimental schedule by referring to previous research results and observed for 14 days after cell injection [29]. For the NIS scoring test, a total of 12 mice were randomly divided into 3 groups: HIE mice that were not administered the cells (HIE, *n* = 4), HIE mice that were administered IV ES-MSCs (HIE + ES-MSC, *n* = 4), and normal mice that were not administered the cells (Normal, *n* = 4). For in vivo live cell tracking, a total of 8 mice were randomly divided into 2 groups: HIE mice that were administered IV ES-MSCs (HIE + ES-MSC, *n* = 4) and normal mice that were administered IV ES-MSCs (Normal + ES-MSC, *n* = 4). For in vivo and ex vivo organ imaging by lapse of time, 15 HIE mice were randomly divided into 5 groups (before infusion (0 h), 1 h, 1 day, 7 days, and 14 days after infusion), with *n* = 3 for each group (Figure 1a).

### 2.5. Behavior Test (Neurologic Impairment Scoring)

One day after hypoxic–ischemic induction, a neurobehavior test was conducted using the Neurologic Impedance Score (NIS) to verify that the models were properly created. Neurologic impairment was scored on a 5-point scale: a score of 0 indicated that there was no neurologic deficit, a score of 1 (failure to extend left forepaw fully) indicated a mild focal neurologic deficit, a score of 2 (circling to the left) indicated a moderate focal neurologic deficit, a score of 3 (falling to the left) indicated a severe focal deficit, and mice with a score of 4 could not walk spontaneously and had a depressed level of consciousness [30]. Mice exhibiting an NIS of at least 2 after hypoxic–ischemic induction were selected for evaluation.

### 2.6. In Vivo and Ex Vivo Imaging

Noninvasive fluorescence imaging was performed using the Pearl Trilogy Small Animal Fluorescence imaging system (LI–COR; Lincoln, NE, USA) at 800 nm representing ES-MSC [31]. We also acquired white light digital images. Fluorescence signal intensities were measured within user-defined regions of interest (ROIs) using Image Studio Software, LI–COR Biosciences. We then calculated the mean fluorescence signal from the co-registered ROIs of each mouse. First, all measurements were normalized to the background signal. Then, each organ fluorescence signal activity was averaged by the number of ROI pixels, which served as a surrogate measure of quantity of ES-MSCs accumulated in each organ. The extremities of the mice were fixed with 3M tape (3M™ Textile Flatback Tape 2526) while scanning to minimize movement that may result in organs shifting and affect FLI values. Live imaging was conducted in both the normal and HIE groups before ES-MSC infusion (0 h) and 1 h, 1 day, 7 days, and 14 days after the infusion (Figure 1a). To account for the inherent differences in organ size, the signal intensity was calculated as infiltrated cell FLI per unit area. Fluorescence was measured with the mouse in the supine position for the liver, lungs, and spleen, and in the prone position for the brain (Appendix A). 

### 2.7. Immunofluorescence Analysis

The hypoxic–ischemic animals treated with ES-MSCs were subjected to deep anesthesia with isoflurane 2 weeks later and their brains removed. After perfusion with PBS, the extracted brains were fixed in 4% paraformaldehyde for 24 h. The fixed brain samples were immersed in 30% sucrose for cryoprotection and cut into 20 µm–thick slices using a Cryostat (Leica, Wetzlar, Germany). Sections were attached to glass slides and blocked with 3% bovine serum albumin (BSA). Then, the sections were immersed in a primary antibody solution. Fluorescent targets were NeuN, to assess neuronal survival, and Human Nuclei (HuNu), as a human cell tracing marker. The brain tissues were incubated overnight at 4 °C with the following primary antibodies: anti-NeuN (1:100, Novus, Littleton, CO, USA) and anti-HuNu (1:100, Abcam, Cambridge, UK). Sections were rinsed with PBS with Tween 20 buffer and soaked in fluorescein isothiocyanate (FITC)–linked secondary antibody (Alexa 488 or 594, 1:1000) for an additional 2 h at room temperature. A coverslip was placed on top of the tissue, and the sample was mounted using 100 μL of VectaShield with DAPI (Vector Laboratories, Inc., CA, USA) for imaging. Then, montage images were captured using Cytation™ 5 (BioTek Instruments, Inc., Winooski, VT, USA), a modular multimode microplate reader coupled with an automated digital microscope. Images were acquired by imaging the whole brain in a 5 × 6 montage [32]. Immunofluorescence imaging was visualized using a NIKON fluorescence microscope (NIKON Instruments, Inc, Melville, NY, USA), and quantification was performed using ImageJ software (National Institutes of Health, US).

### 2.8. Statistical Analysis

Data are presented as the mean ± the standard deviation. Statistical comparisons between normal and HIE groups were performed at 1 h, 1 day, 7 days, and 14 days after infusion using one-way ANOVA analysis with SPSS version 21.0 (IBM, Chicago, IL, USA). A value of *p* < 0.05 was considered statistically significant.

## 3. Results

### 3.1. Improvement in Neurological Behavior in the ES-MSC–Administered HIE Group and Infiltration of the Cells in the Damaged Brain

To determine whether the neurological behavior was improved by ES-MSC administration, NIS scores were measured. The score of NIS was 0 points before HIE induction surgery in all groups, but the score increased significantly after HIE induction (*p* < 0.01). In the group that received ES-MSC, the score decreased from 7 days postinfusion compared with the HIE group, which showed a trend of further decrement until 14 days postinfusion (*p* < 0.01) (Figure 1b).

According to the brain NIR image analysis, fluorescence-labeled ES-MSCs were observed from day 1 and day 7 after infusion on the head area in both normal and HIE mice. However, differences in the visualized fluorescence area were notable between the groups—around the cerebellum in normal mice and at the infarct lesion in HIE mice. Furthermore, on day 14 after infusion, unlike normal mice, which showed reduced FLI in the brain, the HIE mice showed maintained elevated FLI in the brain lesion (Figure 1c,d).

### 3.2. Changes in the Distribution of ES-MSCs in the Host Body by Lapse of Time

In vivo NIR imaging enabled us to assess the distribution of ES-MSCs in each organ over time in live mice (Figure 2a,b). To quantitatively confirm the in vivo imaging result of each organ, the relative signal intensity was compared with the signal intensity of each organ as a control immediately after infusion. The FLI of the lungs was markedly elevated 1 h postinfusion and showed still higher values until day 1 postinfusion compared with the baseline in both Normal and HIE groups (*p* < 0.05). The values gradually decreased thereafter. However, 1 h after infusion, the HIE group exhibited a higher signal intensity than the Normal group in the lungs (*p* < 0.05) (Figure 2c). The FLI of the liver was elevated from 1 h to day 7 postinfusion in both Normal and HIE groups, there being no difference between the two groups, while both groups showed maximal values 1 day postinfusion (Figure 2d). The spleen did not exhibit any fluorescent signals 1 h after infusion. Similar to the liver, it showed the strongest intensity on day 1 after infusion, and this intensity gradually decreased until day 14 after infusion, there being no difference between the groups (Figure 2e). Other notable findings are wide variances in the FLIs of the liver and spleen on day 1 postinfusion in the HIE group.

### 3.3. Ex Vivo NIR Fluorescent Imaging on the Infiltration Ability of Stem Cells for Each Organ over Time

In the same way as in the in vivo image, the intensity was measured by excising the lungs, liver, spleen, and brain of the HIE models injected with ES-MSCs to confirm the amount of cells infiltrated into the actual tissue at the same time points (Figure 3a). In the extracted lungs, the signal was measured from 1 h after infusion. The signal was significantly strong 1 day after infusion and suddenly decreased from days 7 to 14 postinfusion (Figure 3b). Similarly, in the extracted liver, while the penetration of ES-MSCs observed 1 h postinfusion had no statistical significance, the signals were elevated from days 1 to 7 postinfusion (*p* < 0.05). After day 7, the FLI of the liver showed a tendency to decrease with time (Figure 3c). In the spleen, the fluorescent signal was almost absent 1 h after infusion, was strongly elevated on day 1 postinfusion (*p* < 0.01), and rapidly decreased afterward (Figure 3d). In the brain, infiltration into the ischemic damaged area was visualized on day 1 postinfusion, and no statistical difference was observed. The signal became apparent on days 7 and 14 postinfusion (*p* < 0.05), with a tendency to increase in intensity until the last follow-up (Figure 3e).

### 3.4. Engraftment of ES-MSCs in Hypoxic–Ischemic Brain Lesion

Finally, we investigated, using actual brain tissue slices (not indirect observation of the presence or absence of cells through fluorescence), the ability of cells to engraft into the brain. Two weeks after ES-MSC infusion into HIE mice, when the FLI signal was elevated in the brain, the existence of human nuclear cells was determined using HuNu, a marker for human cells. In the brain slices of mice that received ES-MSCs, HuNu (+) cells were observed (Figure 4a). However, the HuNu (+) cells were not found in the tissue of the control group (which did not receive ES-MSCs). The result of cross-staining with NeuN, a marker of neuronal survival, showed small numbers of NeuN (+) cells with an enlarged shape in the HIE group, while these findings representing neuronal damage were not seen in the HIE + ES-MSC group, which had a higher number of surviving neurons (Figure 4b).

## 4. Discussion

Studies have shown the therapeutic potency of stem cells for neurological diseases, such as cerebral ischemia [33]. However, to be used in the clinical setting, their safety and efficacy should be guaranteed first, but these are not yet fully acquired [34]. Moreover, the mode of action of stem cell therapy in brain damage cases has not been fully identified. This action can be more complex in the case of systemic IV administration, the most frequently tried administration method up to now [35]. Accordingly, it is essential to monitor the fate of administered cells in vivo to understand the therapeutic mechanism. In this study, the efficacy of intravenously introduced ES-MSCs in HIE was confirmed by neurobehavioral assessment as the first step. Then, the authors also assessed the distribution of ES-MSCs in the host body that was changed with time.

Since the brain is regarded as the most important organ in ES-MSC therapy for HIE, the finding is described previously. The signal intensity, which is considered to represent the infiltration of ES-MSCs, showed different patterns in HIE mice and noninjured normal mice. In the brains of noninjured normal mice, the FLI increased from the day after ES-MSC infusion, with the main signal located near the cerebellum, apparently at the lambda point of cranial venous sinuses [36]. However, in HIE mice, the signal appeared more intense at the ischemic lesion site and remained intense until 14 days postinfusion. During this time, the brain area of normal mice did not show any fluorescence (Figure 1d). This is in line with the report that hypoxic conditions enhance the homing and neuroprotective abilities of MSCs [37,38]. The findings in normal mice are presumed to demonstrate that ES-MSCs remain in the intravascular area, without penetrating the blood–brain barrier (BBB), while the signal in an ischemic lesion seems to represent the penetration of the cell components across the BBB. Similar research that used neural stem cells reported the localization of the cells at the ischemic lesion after IV administration, which indicates the ability of therapeutic cells to cross the BBB [39]. Cerebral ischemia could have disrupted the BBB, increasing the ability of ES-MSCs to permeate the lesion [40]. In addition, according to recent reports, the signals found in the brain lesion may be MSC-derived extracellular vesicles (EVs), since those carry important cell components, including membrane lipids, where CellVue^®^ attaches for labeling [41].

The sequential migration of the administered ES-MSCs in organs by lapse of time could be monitored also by in vivo NIR imaging. Injected cells were found not only in the brain but also in various organs throughout the body, mainly the lungs, liver, and spleen (Figure 2). One study suggested that stem cells delivered by the IV route could be entrapped in peripheral tissues (lungs, liver, spleen, and kidneys), particularly in the lungs [42], limiting cell delivery to the brain. Small cells such as bone marrow–derived MSCs (7 μm in diameter) have been shown to infiltrate the lungs up to 30 times more than larger stem cell types such as adipose-derived MSCs (18 μm) and neural stem cells (16 μm) [43]. Another study reported that bone marrow–derived MSCs migrated preferentially to the spleen and were detected in the head region between 12 h and 11 days after IV administration [44]. Although the roles of the organs that the cells migrate to are not fully understood, a previous report revealed the importance of the spleen in MSC therapy for cerebral infarct [45]. In addition, it has been suggested that macrophages in the lungs, which ES-MSCs necessarily come in contact with when passing through the capillaries of the lungs, play a significant role in stem cell therapy through polarization [46]. In this study, HIE mice showed higher infiltration of ES-MSCs in the lungs 1 h postinfusion compared with non-injured normal mice. Even though the meaning of entrapment in the lung capillary has not been elucidated, it may be related to therapeutic effect in cell therapy. On day 1 after infusion, the strongest signals in the liver and spleen seem to be related to metabolism, clearance, and inflammation in both the normal and HIE model [29,47]. Brain signal intensity was also elevated from day 1 after infusion, which is thought to be due to the BBB opening caused by hypoxic–ischemic damage, as suggested [48].

In a longitudinal study that tracks cells in vivo, it would be necessary to verify whether the injected cells actually appear in the organs. When the signal intensity in in vivo imaging was compared with that in ex vivo findings for measuring cell distribution in HIE mice, the overall trend of migration was found to be similar over time. However, the lungs and brain showed somewhat different manifestations over time. In the extracted lungs, a strong signal appeared 1 day postinfusion, and the in vivo intensity was the highest 1 h after infusion, with the elevation maintained until 1 day postinfusion. The differences could have been caused by depth from the skin, which means more superficial circulation just after the infusion and entrapment in deeper pulmonary tissues 1 day post-infusion. In a previous study related to stem cell tracking, the most cells were observed in the lungs 1 min after IV administration of stem cells [46]. In another report, most stem cells introduced into a neuronal disease model via IV were found to be sequestrated in the lungs both in vivo and ex vivo 1 day after the cell administration [47]. These results are nonunified and could have been affected by cell types, animal model, and detection equipment. Dissociative findings in the brain of an HIE on day 1 postinfusion, with elevated signal in vivo and nonsignificant finding ex vivo, should have been caused by venous pooling of ES-MSCs in vivo near the cerebellum, as in normal mice. Since the brain signal was taken up from the whole cranial area without dividing the infarct lesion and other areas, intensities of the separate areas were not obtained. However, the signal intensity in the ischemic lesion became stronger over time until the last follow-up, which might have significance.

Finally, to confirm the validity of in vivo results, the engraftment of ES-MSCs in the brain using HuNu, a marker for staining human cells, was assessed. In the brain of HIE mice that received cell therapy, HuNu was observed in the periventricular region. This finding indicates correspondent results of in vivo imaging and infiltration of ES-MSCs into the brain. In addition, the brain tissue of HIE mice that did not receive ES-MSCs showed enlarged neuronal cells, which indicates an early stage of nerve cell death due to ischemic injury, while the brain that had received ES-MSCs showed normal-sized neuronal cells (Figure 4a,b).

This study has several limitations. Although noninvasive live imaging is possible and NIR imaging has a higher penetration depth than other devices, signals from tissues deep in the body cannot be fully obtained, and the resolution gives limited information. In addition, superficial signals can be hindered by the luster of sacrificed animal tissue. Although the signal intensity was regarded as representing the existence of ES-MSCs, and HuNu (+) cells were actually found in the ES-MSC–treated brain, the exact status of the cells cannot be assessed. The possibility of ES-MSC–derived exosomes or cellular particles should be reconsidered. However, compared with other devices, NIR imaging enabled us to monitor the migration of the cells or the components in real time. With concurrent analysis of the cellular role of each organ in which stem cells remained, therapeutic mechanisms of stem cells might be further clarified.

## 5. Conclusions

The findings of this in vivo experiment revealed the homing effect of ES-MSCs in the brain lesion that lasted up to 14 days after infusion, which is informative in understanding the therapeutic efficacy mechanism of ES-MSCs in HIE. The roles of the organs in the action mechanism of systemically administered stem cells should be clarified through further studies.

## Figures and Tables

**Figure 1 life-13-00227-f001:**
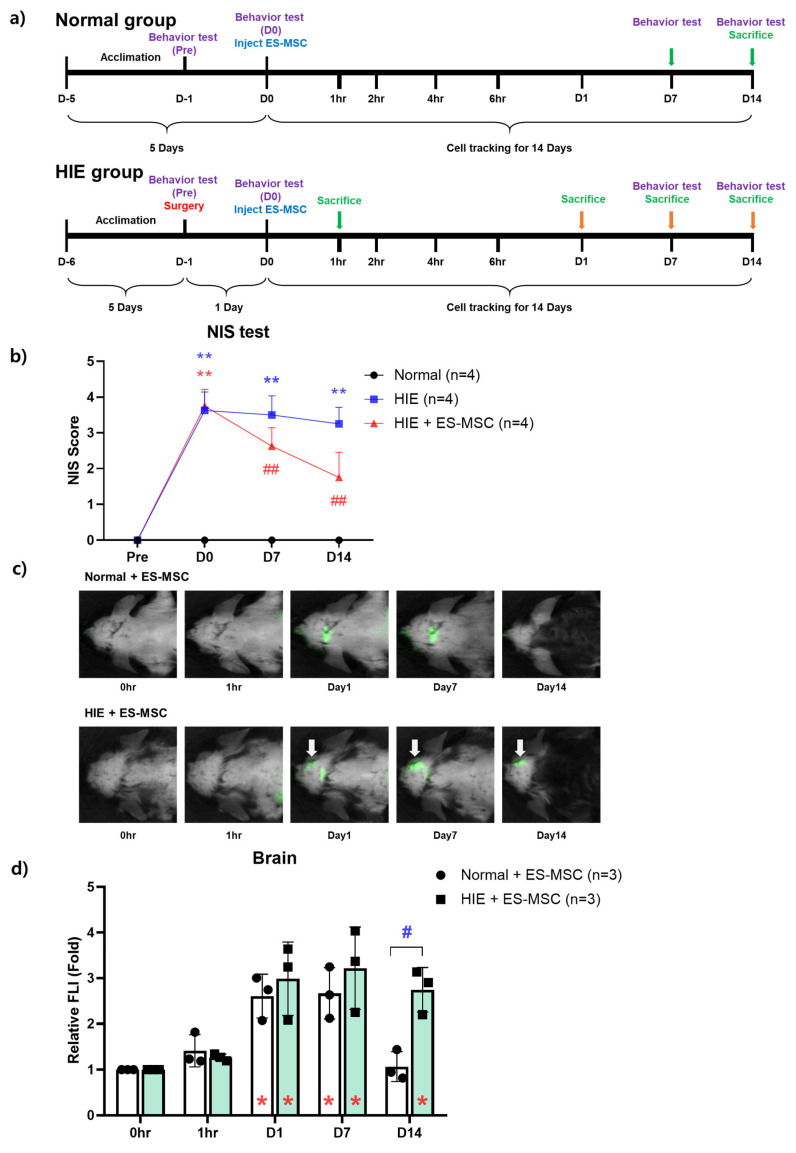
Neurobehavioral improvement of ES-MSCs in an ischemic brain injury model and the maintenance of intracerebral infiltration of ES-MSCs until day 14 after the infusion. (**a**) Experimental schematic and schedules. (**b**) Neurobehavioral evaluation of functional recovery in hypoxic–ischemic encephalopathy mice after ES-MSC infusion. (**c**) Engraftment ability in the brain over time after the administration of ES-MSCs to normal mice and hypoxic–ischemic mice. (**d**) Determination and quantification of the fluorescence intensity. Data represent the mean ± the standard deviation. NIS stands for Neurologic Impedance Score. The significance of difference is marked as * *p* < 0.05, ** *p* < 0.01 vs. each baseline (Pre) score, and # *p* < 0.05, ## *p* < 0.01 vs. Normal control group for both NIS and relative fluorescence signal intensity (FLI) data.

**Figure 2 life-13-00227-f002:**
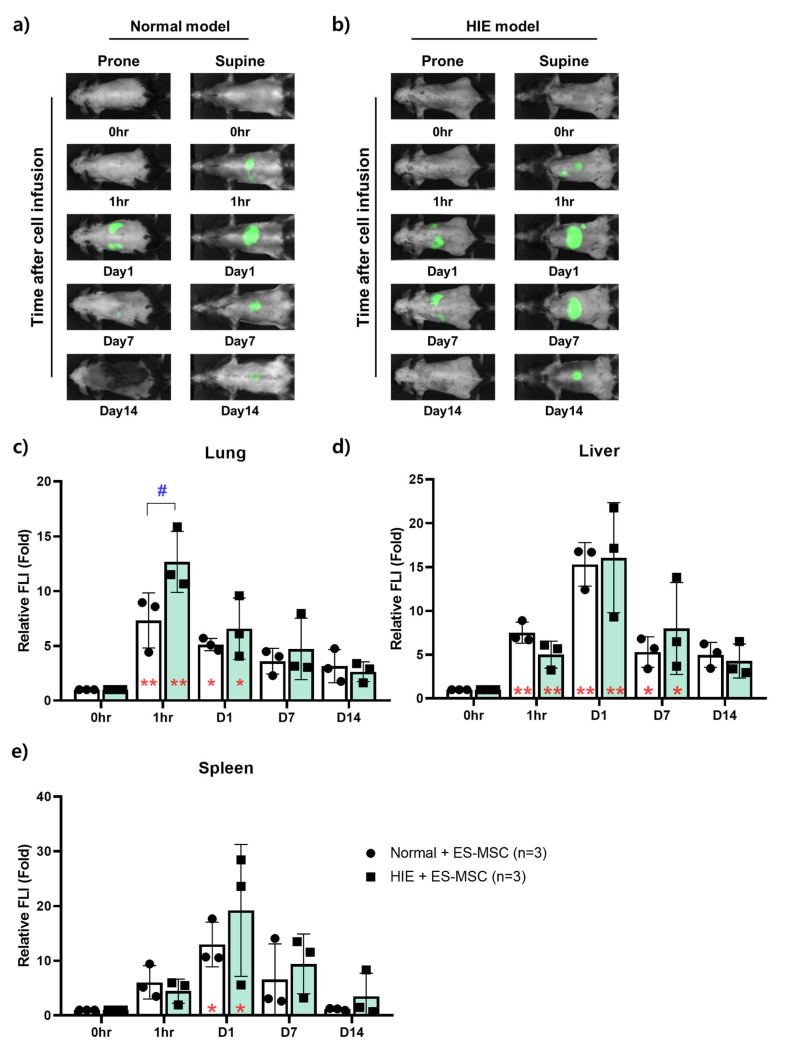
Infiltration capacity of cells in organs over time. Intra-organ distribution of infused stem cells over time in (**a**) normal and (**b**) hypoxic–ischemic encephalopathy models. Time-dependent tracking of the distribution of administered stem cells in the (**c**) lungs, (**d**) liver, and (**e**) spleen in normal or hypoxic–ischemic encephalopathy models. Data represent the mean ± the standard deviation. The significance of difference is marked as * *p* < 0.05, ** *p* < 0.01 vs. each baseline (Pre) score, and # *p* < 0.05 vs. Normal control group for relative fluorescence signal intensity (FLI) data.

**Figure 3 life-13-00227-f003:**
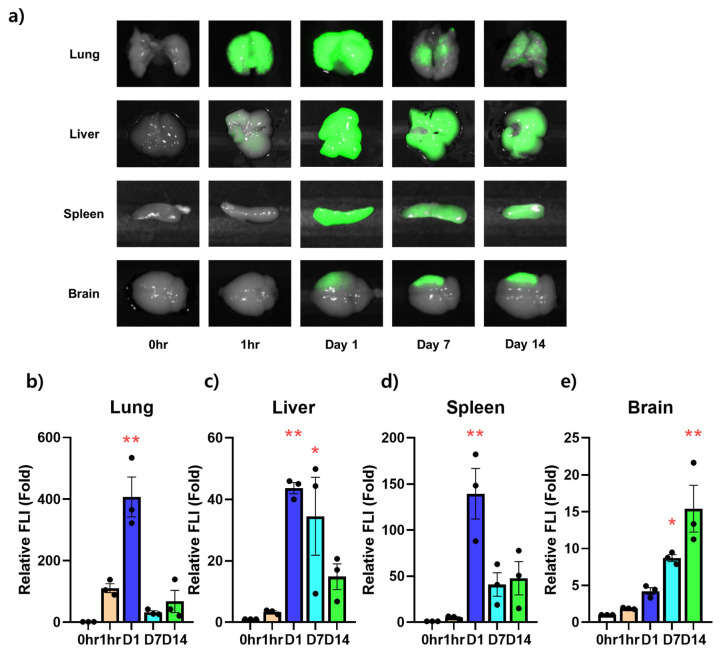
Ex vivo experiment on the time-dependent infiltration ability of stem cells into each organ. (**a**) Images of lungs, livers, spleens, and brains at different times in hypoxic–ischemic animal models. (**b**) Fluorescence intensity analysis of images of lungs, (**c**) livers, (**d**) spleens, and (**e**) brains excised over time in a hypoxic–ischemic animal model. Data represent the mean ± the standard deviation (SD). The significance of difference is marked as * *p* < 0.05 and ** *p* < 0.01 vs. each baseline (Pre) score for relative fluorescence signal intensity (FLI) data. *n* = 3 per group.

**Figure 4 life-13-00227-f004:**
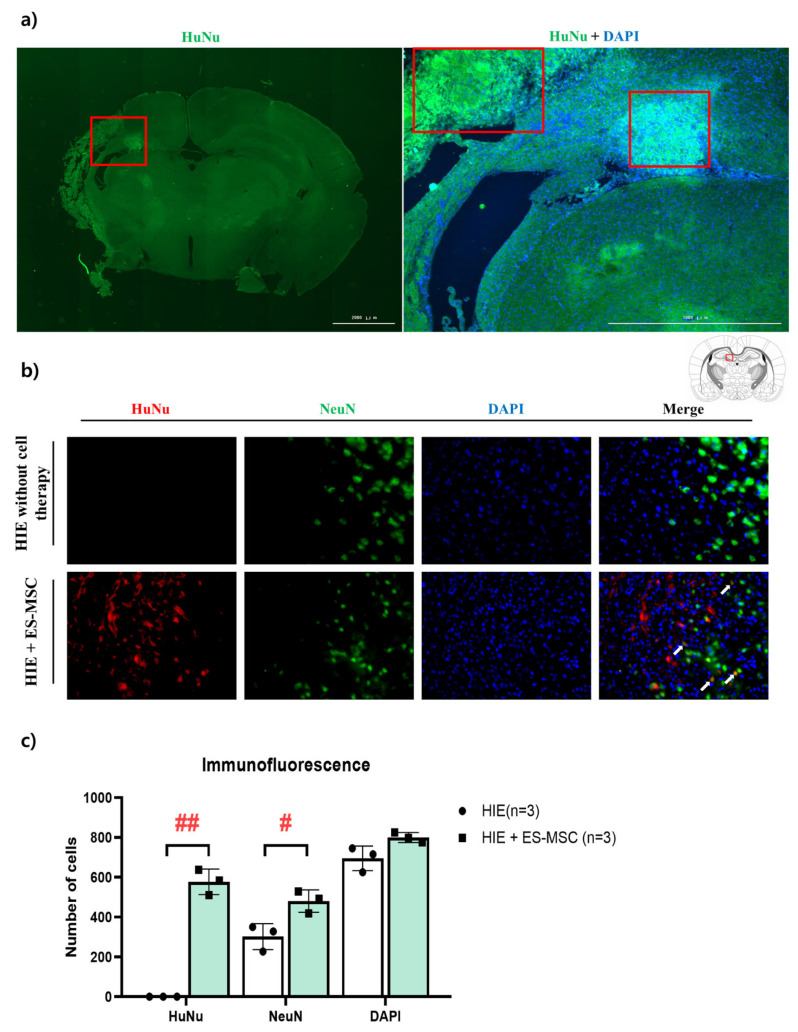
Immunofluorescence analysis of ES-MSC infiltration in the brain in an ischemic–hypoxic model. (**a**) Montage image (40× magnification) showing variability in HuNu (human nuclei marker; green) and DAPI (pan-nuclear DNA stain; blue) expression labeling across the whole mouse brain tissue slice. (**b**) Immunofluorescence images (200× magnification) of ischemic brain immunolabeled for HuNu (red), NeuN (green), and DAPI (blue). Arrow (white) means HuNu/NeuN double-positive cells. (**c**) HuNu-, NeuN-, and DAPI-positive cells quantified using ImageJ software. Data represent the mean ± the standard deviation. The significance of difference is marked as # *p* < 0.05 and ## *p* < 0.01 vs. each HIE control group.

## Data Availability

The data presented in this study are available from the corresponding author upon request.

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
