# Peer review of "Distribution of Embryonic Stem Cell-Derived Mesenchymal Stem Cells after Intravenous Infusion in Hypoxic–Ischemic Encephalopathy"

_life, 2023, doi:10.3390/life13010227_

Round 1
Reviewer 1 Report
This manuscript by Lee et al is overall well written. The experiment design is thorough although animal numbers are relatively fewer, given it is time sensitive, it is acceptable. The authors used HIE model to test the ES-MSC distribution in the brain and other organs. They found that IV ES-MSCs are mainly distributed in the target organ brain as well as Lung, liver, and spleen. The results are convincing and presented clearly.
Following are some comments to further improve the manuscript:
Summary: Line 21, Introduced cells meaning obscure, it is better replaced with “Infused cells” since authors performed IV infusion.
Lines 25-26: Conclusion sentence not specific, may be replaced with: In conclusion, the home effects and involved organs could have exerted therapeutic effects on the hypoxic-ischemic brain damage.
Abstract: Last sentence is not clear, did the authors mean the migration of infused cells into other organs play a role in brain damage?
Methods: Line 101, why mice are individually housed?
NIS scoring: N=4 relatively few animals but it is understandable, fluorescent tracking is time sensitive.
The authors injected human ES-MSCs into normal C56BL/6J mice, there will be definitely immune rejection which can affect the results. Why the authors did not choose nude mice or SCID mice? Does Human ES-MSC have immune privilege?
Line 129, For the in vivo time-lapse imaging, did the author only used HIE group injected with ES-MSCs? This is the case from the results presented. Then the authors should state “15 HIE mice” instead of “15 mice”.
Results: Line 191, did the HIE+ES-MSC group show significant NIS score decrease compared to HIE group only at day 7 and day 14? They looked different. These will be important results to show the improvement of ES-MSC injection on HIE-induced brain damage.
Figure 1 C legend, why the author said invasive ability over time, it is obviously the Non-invasive time- lapse in vivo fluorescent images for the cell distribution in the brain. Please revise.
Figure 1 legend, line 206, why the authors stated: N=8 while they labeled N=4 or N=3 in the figures. Please correct. It is not necessary to state N here because they already have it on the figure and methods section.
Figure 2, a,b, “categorize” better be replaced with “Time after cell infusion” to be more straightforward.
Line 213: “Abruptly” maybe replaced with “sharply” or “markedly”.
Line 233 subtitle: “Ex vivo experiment” maybe better replaced with “Ex vivo NIR fluorescent imaging” to be more specific.
Line 257 Subtitle: “Invasion” better be replaced with “Engraftment” because invasion tends to be used for tumor. Engraftment is a common word used in stem cells to describe cells’ contribution to tissue repair.
Line 260, remove “brain”, this is redundant word use.
Figure 4: Merged images of HIE+ES-MSC group appeared to have HuNu/NeuN colocalization which may indicate the infused-ES-MSC differentiated into neurons. Suggest using arrows to point this. Many other cells are not HuNu/NeuN double positive. Using insets to highlight double-positive cells will be also helpful.
Figure 4 C: “Immunohistochemistry” Should be “Immunofluorescence” to be more accurate.
Reviewer 2 Report
1. oxygen, because of which brain neurons are damaged, which leads to serious aftereffects --- English needs to be corrected.
2. Line 51: the lost tissues
3. Line 56: and might--- and may
4. Line 82: organ? You mean injected organ or target organ?
5. Line 84: using NIR--------by using NIR
6. Line 85:might? ------may
7. Experiment design: how did you decide the time point for the cell tracking is the “14 day” would it be possible that the NIS scores will be better after the 14 day, or worse after the day?
8. Why the values of FLI on the 14 day were lower than the 7th day? (Figure 1 d)
9. Figure 3 (e): It seems to me that the FLI of the brain on D7 is similar to that on D14
10: line 278-287: English is difficult to be understood.
11. Your HIE model is interesting. However, in addition to the binding of the right carotid artery, did you consider that the Isoflurane you used in the study may also affect the rat’s brain (please see NeuroImage Volume 234, 1 July 2021, 117987 ; https://doi.org/10.1016/j.neuroimage.2021.117987 For further information). Therefore, would it be possible that the sequence of your stem cells showing in different organs may be due to the isofluran’s effects?
12. The meaning of the time points for stem cell migrating to different organ need to be carefully interpreted.
13. English needs to be completely corrected by a native speaker.
Round 2
Reviewer 2 Report
The reviewer is happy with The authors' responses. I would like to recommend the journal to accept it.